# Capsid structure of a fungal dsRNA megabirnavirus reveals its previously unidentified surface architecture

Han Wang[1], Lakha Salaipeth[2¤], Naoyuki Miyazaki[3]*, Nobuhiro Suzuki[2]*, Kenta Okamoto[1]*

1 The Laboratory of Molecular Biophysics, Department of Cell and Molecular Biology, Uppsala University, Uppsala, Sweden, 2 Institute of Plant Science and Resources, Okayama University, Kurashiki, Okayama, Japan, 3 Life Science Center of Survival Dynamics, Tsukuba Advanced Research Alliance, University of Tsukuba, Tsukuba, Ibaraki, Japan

¤ Current address: School of Bioresources and Technology, King Mongkut's University of Technology Thonburi, Bangkok, Thailand

* naomiyazaki@gmail.com (NM); nsuzuki@okayama-u.ac.jp (NS); kenta.okamoto@icm.uu.se (KO)

**Data Availability Statement:** The icosahedral full and empty maps and symmetry-expanded cryo-EM maps are available in the EMDB repository, entries EMD-15855, EMD-15857 and EMD-15859, respectively. The atomic models of the MCPs and

## Abstract

Rosellinia necatrix megabirnavirus 1-W779 (RnMBV1) is a non-enveloped icosahedral double-stranded (ds)RNA virus that infects the ascomycete fungus *Rosellinia necatrix*, a causative agent that induces a lethal plant disease white root rot. Herein, we have first resolved the atomic structure of the RnMBV1 capsid at 3.2 Å resolution using cryo-electron microscopy (cryo-EM) single-particle analysis. Compared with other non-enveloped icosahedral dsRNA viruses, the RnMBV1 capsid protein structure exhibits an extra-long C-terminal arm and a surface protrusion domain. In addition, the previously unrecognized crown proteins are identified in a symmetry-expanded cryo-EM model and are present over the 3-fold axes. These exclusive structural features of the RnMBV1 capsid could have been acquired for playing essential roles in transmission and/or particle assembly of the megabirnaviruses. Our findings, therefore, will reinforce the understanding of how the structural and molecular machineries of the megabirnaviruses influence the virulence of the disease-related ascomycete fungus.

## Author summary

A fungal plant soil-borne pathogen, *Rosellinia necatrix*, which can cause devastating disease white root rot in many highly valued fruit trees, is difficult to be controlled with conventional approaches such as fungicide applications. Rosellinia necatrix megabirnavirus 1-W779 (RnMBV1) is a dsRNA virus isolated from the *R. necatrix* field strain, W779, and this virus can be a viro-control candidate to confer hypovirulence in its host *R. necatrix*. To make use of RnMBV1 in the white root rot disease control, more molecular and structural investigations will offer us more insights. Here, we have performed cryo-electron microscopy (cryo-EM) single-particle analysis, to obtain the first atomic models of RnMBV1 particles. Based on the atomic structures, we found unique both surface and

CrPs are available in the PDB repository, entries 8B4Z and 8B59.

**Funding:** Funding was provided by the following agencies: Vetenskapsrådet (VR)/The Swedish Research Council (to K.O., grant no. 2018-03387), FORMAS research grant from the Swedish Research Council for Environment, Agricultural Sciences, and Spatial Planning (to K.O., grant no. 2018-00421), and the Royal Swedish Academy of Sciences (to K.O., grant no. BS2018-0053). Grants-in-Aid for Scientific Research on Innovative Areas from the Japanese Ministry of Education, Culture, Sports, Science and Technology (KAKENHI 21H05035, 17H01463, 16H06436, 16H06429 and 16K21723 to N.S., 15K18521 and 18K06154 to N.M.), the Collaborative Study Program of the National Institute for Physiological Science (to N.M.), the Platform Project for Drug Discovery, Informatics, and Structural Life Science (PDIS) from the Ministry of Education, Culture, Sports, Science and Technology (MEXT) and for Supporting Drug Discovery and Life Science Research (Basis for Supporting Innovative Drug Discovery and Life Science Research (BINDS)) (to N.M.). The funders had no role in study design, data collection and analysis, decision to publish, or preparation of the manuscript.

**Competing interests:** The authors have declared that no competing interests exist.

interior features. In addition, we found a previously unidentified protein on the viral surface. These aforementioned structural features might play important roles in the viral life cycles, and will encourage us to apply this fungal virus as a viro-control approach.

## Introduction

Double-stranded (ds)RNA viruses infect a broad spectrum of hosts and significantly impact fisheries, agriculture, food manufacturing, animal welfare, and human health [1–3]. These viruses have been categorized into 12 families—*Birnaviridae*, *Partitiviridae*, *Curvulaviridae*, *Amalgaviridae*, *Picobirnaviridae*, *Cystoviridae*, *Chrysoviridae*, *Totiviridae*, *Quadriviridae*, *Spinareoviridae*, *Sedoreoviridae*, and *Megabirnaviridae*—and a genus *Botybirnavirus*, whose family remains unassigned [4,5]. All dsRNA viruses, excluding birnaviruses, belong to either the phylum *Duplornaviricota* or phylum *Pisuviricota*. The family *Birnaviridae* remains unassigned to any order or phylum. According to the literature, most dsRNA viruses rely on general structural features regarding infection, replication, and proliferation strategies within their life cycles [3,6,7]. All these families, excluding *Birnaviridae*, exhibit a conserved capsid shell structure with the icosahedral symmetry $T = 1$ encompassing the viral genome [7–9]. In contrast, birnavirus exhibits an icosahedral capsid structure with $T = 13$ symmetry [9,10]. In addition, *Cystoviridae*, *Spinareoviridae*, and *Sedoreoviridae* viruses have a $T = 1$ inner capsid shell and an evolutionary acquired $T = 13$ outer capsid shells [11,12]. The conserved capsid shell of the icosahedral $T = 1$ dsRNA viruses comprises 120 chemically identical capsid proteins (CPs) with $\alpha$-helix-rich $\alpha + \beta$-fold structures [3,8], with the $T = 1$ capsid lattice and the conserved structural folds of CPs being their principal structures.

The dsRNA viruses can infect unicellular hosts, such as yeast and protozoa [13–15] and multicellular hosts (metazoa) [1,6,16,17]. Unlike yeast/protozoan dsRNA viruses, metazoan viruses must have acquired both genomic and structural features, which play key roles in cell adhesion and entry mechanisms [18]. All members of the family *Totiviridae* infect yeast/protozoan hosts, while the viruses with toti-like RNA-dependent RNA polymerase (RdRp) genomes (toti-like viruses) have been increasingly characterized to infect metazoan hosts [1,16–20]. The toti-like viruses share low or modest levels of RdRp sequence identity with totiviruses, and/or have a two-open-reading-frame (ORF) genome segment encoding major capsid protein (MCP) and RdRp. Unlike totiviruses [21,22], metazoan toti-like viruses feature previously unrecognized, protruded proteins atop the 5-fold axes of the capsid surface [23,24]. Metazoan *Picobirnaviridae* viruses have acquired capsid protrusion arches/loops, which may provide insertion sites, functioning in cell entry of their lifecycles or for particles stabilizations [25,26]. *Reovirales* viruses, which infect various metazoan hosts, have acquired 9–12 multi-segmented linear dsRNA genomes and multilayers of capsids for enhancing their infecting capabilities [27]. The mammalian and avian reoviruses from the genus *Orthoreoviruses* under the family *Spinareoviridae* have additional surface spikes, fibers, or turrets as their distinguishing features and possible receptor-binding determinants [18,28]. Some metazoan dsRNA viruses might have acquired surface features for another reason. One fungus-infecting member of the family *Partitiviridae* has a surface arch, possibly for enhancing capsid stability and assembly [29], while in another plant partitivirus, the disordered protrusion apex might serve as a permissive site for enabling symbiotic relationships between viruses and hosts [30].

The family *Megabirnaviridae* has recently been characterized and recognized as dsRNA mycoviruses (fungal viruses) [31]. Rosellinia necatrix megabirnavirus 1-W779 (RnMBV1), derived from the white root rot fungus *Rosellinia necatrix* strain W779, is the exemplar strain

that is classified into the family *Megabirnaviridae* [32–34]. However, an increasing number of possible members of the *Megabirnaviridae* family have been reported from filamentous fungi and oomycetes. Two other unassigned members, Sclerotinia sclerotiorum megabirnavirus 1 (SsMBV1) and Rosellinia necatrix megabirnavirus 2 (RnMBV2), have also been isolated [35,36]. The fungus *R. necatrix* is an ascomycete infecting a broad range of plant hosts especially fruit trees and leading to a lethal disastrous disease, white root rot [32,37]. RnMBV1 is a potential viro-control approach agent for restraining and controlling the white root rot disease, because the virus can confer hypovirulence to its host fungus [3,38].

RnMBV1 is an approximately 52 nm bipartite dsRNA mycovirus composed of dsRNA-1 (8.9 kbp in length, AB512282) and dsRNA-2 (7.2 kbp in length, AB512283) [31,32]. Each dsRNA segment has two ORFs. The dsRNA-1 encodes ORF1 MCP (1240 amino acids in length) and ORF2 RdRp [3]. The RdRp can be expressed as an MCP-RdRp fusion product through -1 ribosomal frameshifting [39]. The dsRNA-2 encodes ORF-3 and ORF-4 functionally unknown protein products [3]. We previously showed a cryo-electron microscopy (cryo-EM) model of the RnMBV1 particles at a low resolution (15.7 Å) [3]; however, the fine structural features of this virus remain unclear. Therefore, this study determines the first atomic structure of the RnMBV1 capsid modeled in the icosahedral and symmetry-expanded cryo-EM models, demonstrating the unique structural features of the extra-long C-terminal arm and protrusion domain in the RnMBV1 MCP and previously unidentified crown proteins (CrPs) encoded by the near-C-terminal portion of ORF3. To the best of our knowledge, these exclusive structural features may function importantly in the megabirnavirus life cycles.

## Results and discussions

### Summary of structure determination and overall capsid structure

According to the cryo-EM micrographs and the 2D averaged classifications, both full and empty particles were observed to have morphologically similar capsids (S1D and S1E Fig). The 3D reconstructions of the full and empty RnMBV1 particles were performed with 12,230 full or 14,602 empty particles, and were then generated with the icosahedral (I) input symmetry. Finally, both maps were generated at an overall resolution of 3.2 Å using the "gold-standard" Fourier shell correlation (FSC) = 0.143 criterion (S1A and S1F Fig). In addition, the 3D reconstruction of the CrPs was obtained by symmetry expansion with 244,609 particle orientations and imposed with C1 symmetry, which generated the final map at 3.3 Å resolution (S1A and S1B Fig). The RnMBV1 MCPs are aligned with a typical $T = 1$ icosahedral lattice (Fig 1). The RnMBV1 capsid is composed of 120 chemically identical MCP subunits and additional surface CrPs (Fig 1). There are overall 60 asymmetric dimers, each of which comprises an A subunit (MCP_A, yellow in Fig 1) and a B subunit (MCP_B, purple in Fig 1). Each MCP_A proximally surrounds the 5-fold axis, while MCP_B is located distally from the 5-fold and otherwise encompasses the 3-fold capsid intermediates (Fig 1). The CrPs sit over the MCP_Bs at the 3-fold axes, and each CrP is composed as a trimer (red, orange, and dark yellow in Fig 1). The intact capsid is approximately 535 Å in diameter, including CrPs, which is larger than most known $T = 1$ encapsidated icosahedral viruses [3]. The amino acid residues Ser2–Lys1193 were modeled for MCP_A, while Ser11–Val1236 were assigned for MCP_B in accordance with the reconstructed full-particle cryo-EM map. The last four residues (Glu1237–Asn1240) were invisible, probably due to their high mobility and heterogeneity (S3C Fig). For each MCP_A, more residues (Thr1194–Val1236) were absent on the C-terminus (Fig 2A). The CrP-trimer was modeled with the residues Lys1292–Glu1426, which originated from a portion of ORF3 in dsRNA-2. The structures of MCP_A and MCP_B were highly similar (Root Mean Square Deviation of all atoms = 1.258 Å) apart from the extended C-terminal arms (Fig 2A).

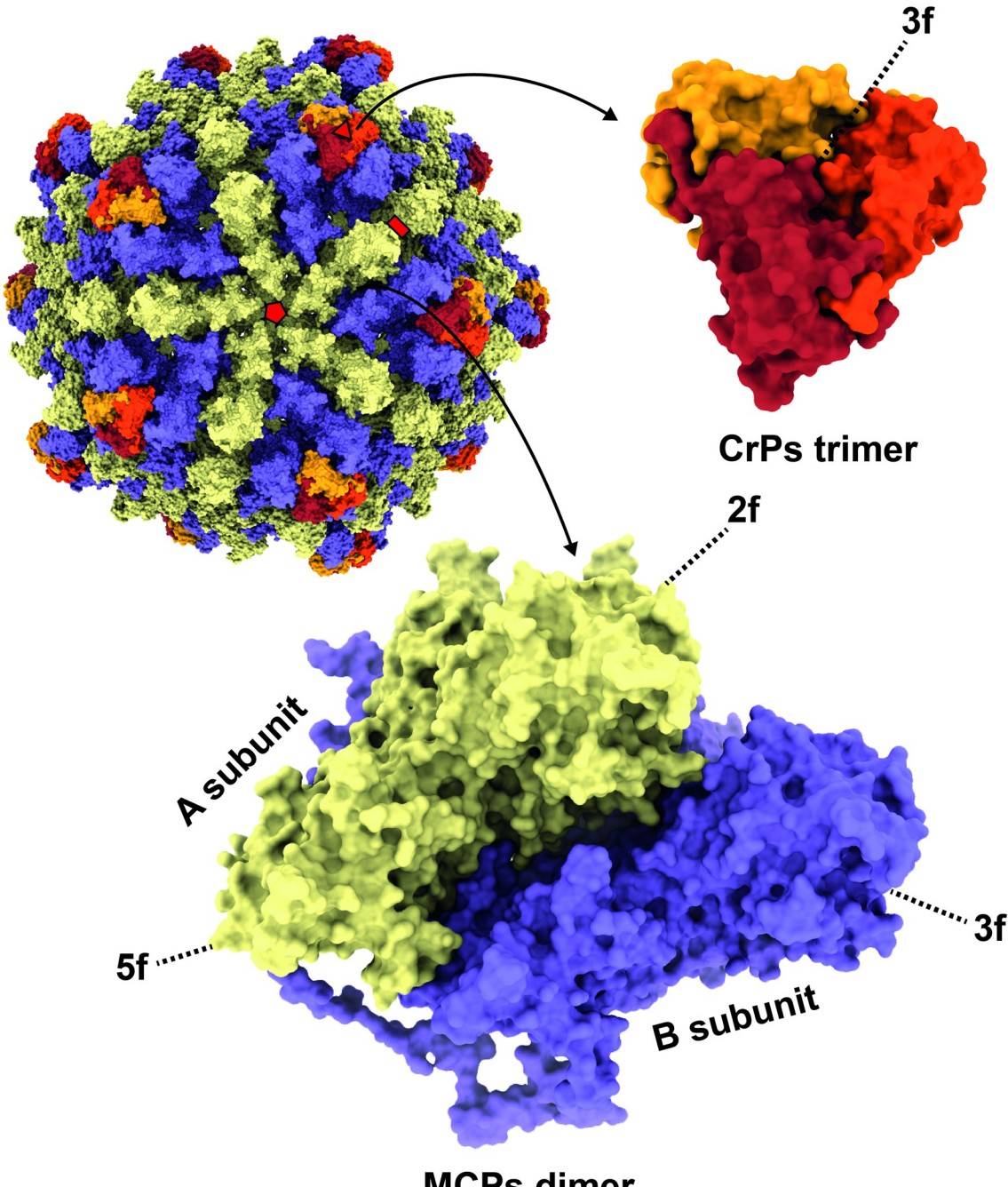

**Fig 1. Overall capsid structure of RnMBV1.** The RnMBV1 capsid comprises 60 MCP dimers (yellow: MCP_A, purple: MCP_B) and the CrP timers (in red, orange, and dark yellow). The red pentagon, triangle, and rectangle show the icosahedral 5-fold (5f), 3-fold (3f), and 2-fold (2f) axes, respectively.

### Interlocking extra-long C-terminal arms of MCPs

The C-terminal arm on each subunit orients differently (Fig 2A) and interacts with one or more adjacent subunits (Fig 2B–2D). In each MCP_B of the RnMBV1 capsid, there is an extra-long 111-amino-acid-residue (Leu1126–Val1236) C-terminal arm/loop, which goes across, interacts, and interlocks with three adjacent subunits in the order of MCP_B1 to A2,

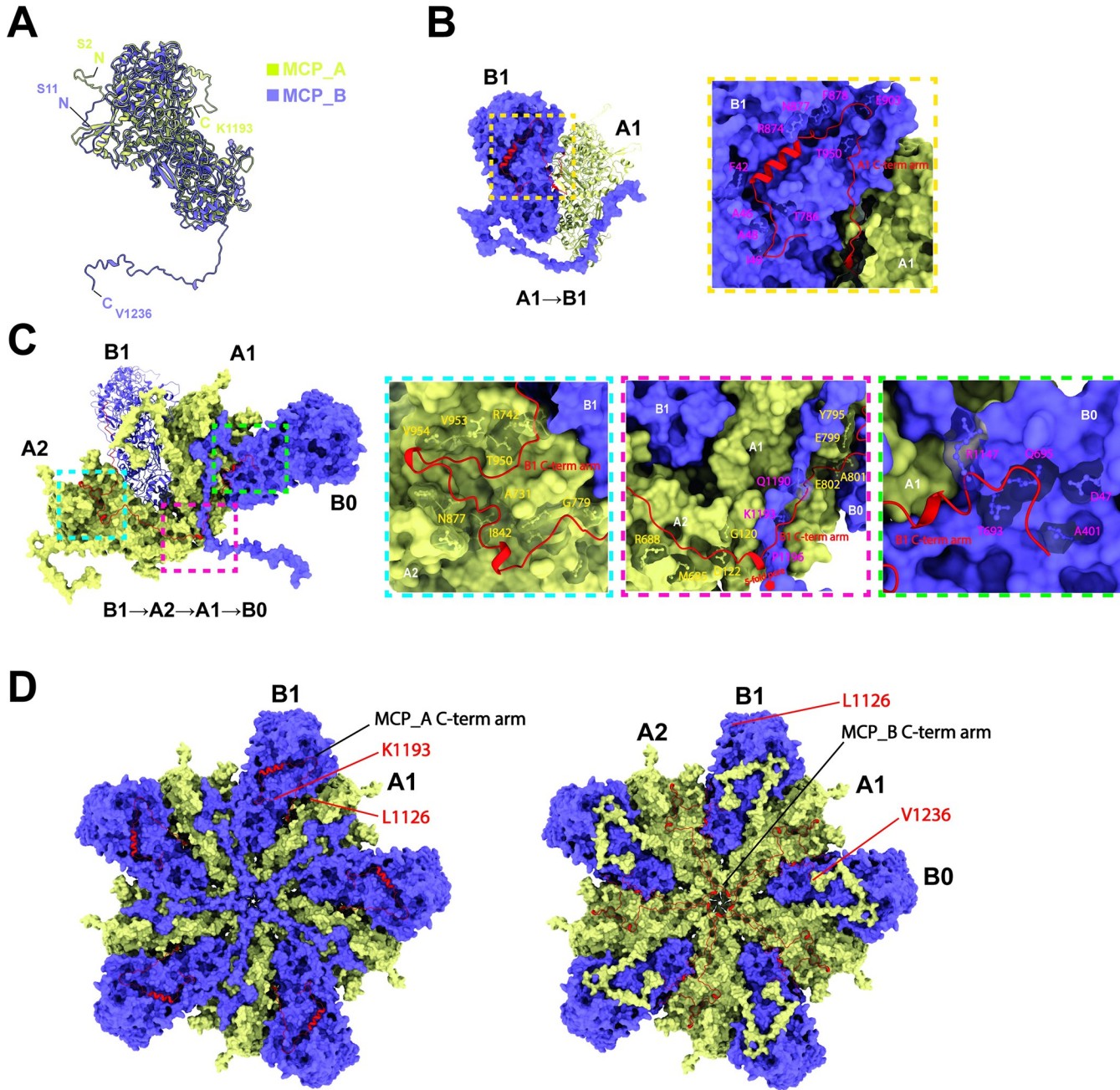

**Fig 2. Extra-long C-terminal arms and the interlocking interface.** A) Superimposition of the MCP_A subunit (yellow) and B subunit (purple). B) Interactions of the C-terminal arm of MCP_A subunit (A1) with one adjacent MCP_B subunit (B1). Close-up view of the interactions between the C-terminal arm of A1 (Leu1126–Lys1193) and the amino acid residues in B1 (dotted yellow box). C) Interactions of the C-terminal arm of MCP_B subunit (B1) with two adjacent MCP_A subunits (A1, A2) and another MCP_B subunit (B0). Close-up view of the interactions between the C-terminal arm of B1 (Leu1126–Val1236) and the amino acid residues in the A2 (dotted light blue box), A1 (dotted red box), and B0 (dotted green box). D) Overall representation of C-terminal arms within a 5-fold complex made up with ten subunits. Both panels show the view inside the capsid. B, C, D) MCP_A and MCP_B subunits are colored yellow and purple, respectively, and are shown in surface representation. The C-terminal arms are colored by red and shown in cartoon representation. The interacting residues are found by contact distance restrictions of 3.8 Å in UCSF Chimera X. The interacting amino acid residues of the MCP_A and MCP_B subunits are labeled in yellow and magenta and represented in ball-and-stick mode.

A1, B0 (Fig 2C and 2D), whereas that of MCP_A (Leu1126–Lys1193) interacts with only one adjacent subunit (Fig 2B and 2D). The amino acid residues of the adjacent MCP_A and MCP_B subunits interacting with the C-terminal arms are displayed in dashed boxes (Fig 2B and 2C). For each MCP_A, the interface (BaveSAS) of A1 to B1 is $5.9 \times 10^3$ Å$^2$, with a shorter C-terminal arm of one $\alpha$-helix (Thr1169–Met1180) lying in the cave of an intra-dimeric MCP_B surface (Fig 2B), while the BaveSAS shrinking to $3.2 \times 10^3$ Å$^2$ (44.8% less interface area) with the MCP_A C-terminal arm deletion in Chimera. For each MCP_B, from the residue Lys1143 to the corresponding C-terminal end, the arm in one B1 starts embedding and meandering along ravines composed of A2, A1, and another B0 subunits, sequentially (Fig 2C and 2D). The BaveSAS surrounding B1 as the interface with A2, A1, and B0 is $11.5 \times 10^3$ Å$^2$, while by performing with the C-terminal arm in-silico deletion, the BaveSAS shrinks down to $6.6 \times 10^3$ Å$^2$ (42.6% less interface area). This indicates that the C-terminal arms contribute to large proportions of the contact interface formation.

Each RnMBV1 MCP has uniquely acquired these multiple interlocking features, each involving up to four subunits utilizing C-terminal extensions. Such a C-terminal extension shares the commonalities with some previously reported C- or N-terminal extension in toti-like viruses (omono river virus, OmRV) [8], *Quadriviridae* (rosellinia. necatrix quadrivirus 1, RnQV1) [40], and *Sedoreoviridae* (rice dwarf virus, RDV) [41]. OmRV CPs display C-terminal arms for interlocking each of the two adjacent subunits [8]. RnQV1 shows multiple hooks/loops/$\alpha$-helices along with inter- and intra-dimer structural units embedded between P2 and P4 CPs [40]. RDV also shows an elongated arm on the N-terminus in favor of tight connections between CPs [41]. However, the CP of the yeast Saccharomyces cerevisiae virus L-A (ScV-L-A) does not show visible C- or N-terminal extension [21]. In dsDNA viruses, the capsid lateral stresses induced by the internal pressure of the genome packaging can be resisted by increased hydrophobic lasso-like intra-subunit interactions that are mediated by several loops and extended arms [42,43]. As distinct from the small ScV-L-A particles and the short genome, the viruses having large particle sizes due to the segmented or longer genomes possibly need interlocking structures to enable stable particle formation and secure genome packaging. In some icosahedral dsRNA viruses, including megabirnaviruses, the C-terminal arm of a few CPs is fused with an RdRp and the CP-RdRp complex is localized within the capsid as a minor structural component during particle assembly [44,45]; therefore, it is interesting to speculate that the RnMBV1 C-terminal arms may play important roles in viral assembly together with the CPs and the fused CP-RdRps.

## Unique protrusion domain of MCP

The obtained atomic model of the RnMBV1 MCP was submitted to the Dali server, and structurally similar CPs in other dsRNA viruses were found (Z-score = 4.7–10.2) (S2A Fig). These dsRNA viruses include Penicillium chrysogenum virus, ScV-L-A, OmRV, RnQV1, Trichomonas vaginalis double-stranded RNA virus 2 (TVV2), and Leishmania RNA virus (LRV1). These matched CP structures were further intensively aligned and compared (S2B Fig). These CP structures are well-aligned on the $\alpha$-helix-rich $\alpha + \beta$-fold conserved domain. As an exception, megabirnavirus RnMBV1 MCP shows an extra protrusion domain (Figs 3, S2C and S2D). The protrusion domains are absent in yeast or protozoan *Totiviridae* and *Chrysoviridae* viruses, such as the totivirus ScV-L-A (Figs 3B and S2D). The virus RnQV1 (family *Quadriviridae*), which infects a multicellular fungus, shows an extra protrusion domain on its MCP (S2C and S2D Fig) [40]. Each full protrusion domain in the RnMBV1 MCP is composed of two fragments of the amino acid residues Ala433–Leu648 and Met965–Ala1125, which form 7 $\beta$-strands (S$_{P-A-G}$) and 13 $\alpha$-helices (H$_{P-A-M}$) (S3A–S3C Fig). The protrusion domains of MCP_A

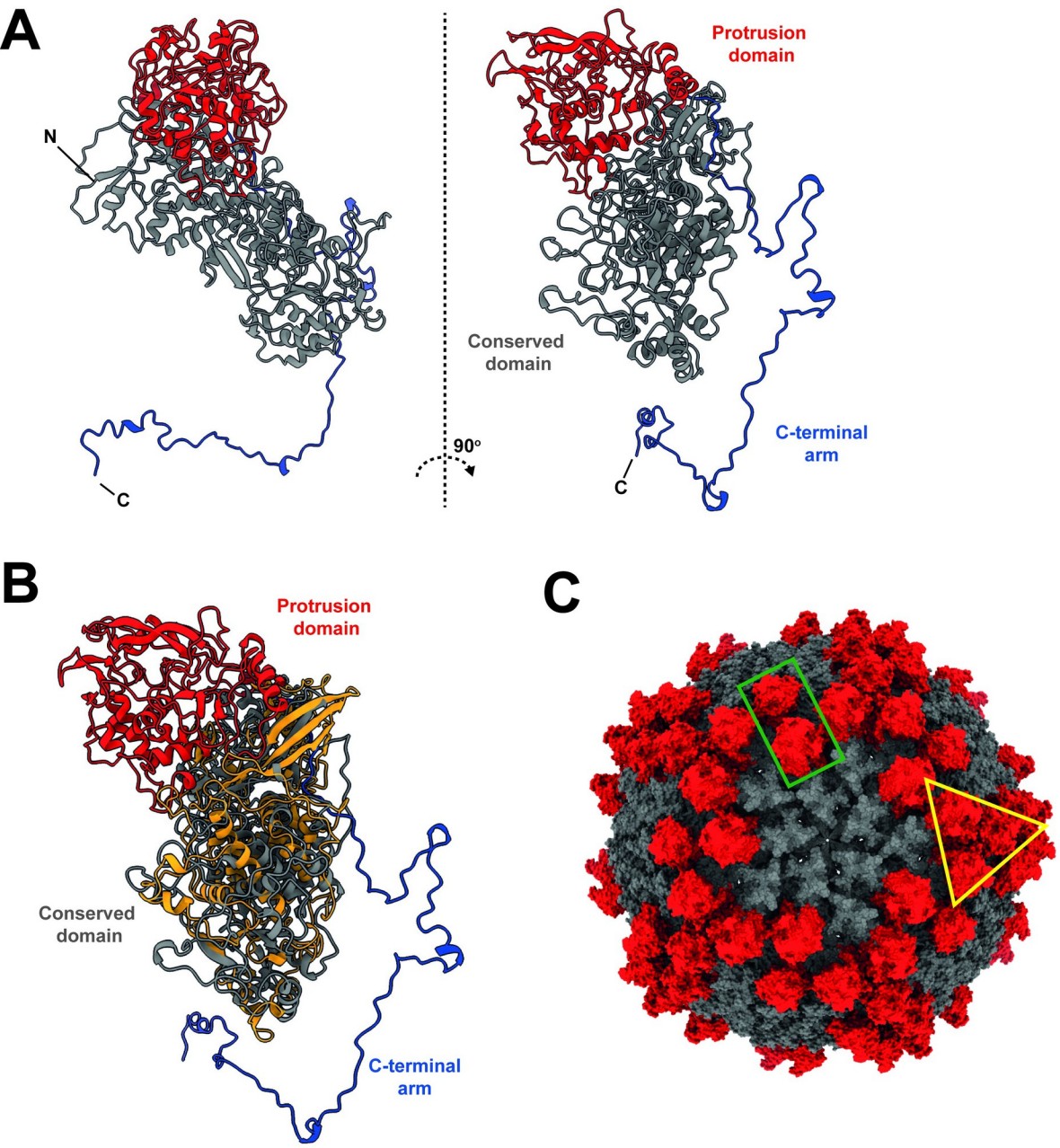

**Fig 3. Unique protrusion domain of the RnMBV1 MCP.** A) Domain organization in the atomic model of the MCP. The protrusion domain, conserved domain, and extra-long C-terminal arm are colored red, gray, and dark blue, respectively. B) Superimposition of RnMBV1 MCP (colored by the domains corresponding to panel A) and ScV-L-A CP (orange). C) Locations of the protrusion domains on the capsid. The protrusion domains and conserved domains are colored red and gray, respectively. The green rectangle and yellow triangle display the protrusion domains formed by the MCP_A subunit dimer and the MCP_B subunit trimer, respectively.

subunits form dimers on the capsid surface (green rectangle in Fig 3C), while those of MCP_B subunits cluster to build a trimer around each 3-fold axis (yellow triangle in Fig 3C).

Such structural acquisitions of the protrusion domains seem to occur independently and uniquely in viruses which infect multicellular hosts in each family among toti-like virus OmRV [6,8], RnQV1 (*Quadriviridae*) [40], and RnMBV1 (*Megabirnaviridae*). The mosquito virus OmRV likely has acquired the protruded surface for an extracellular transmission, such

as receptor binding and membrane penetration [8], whereas neither the multi-segmented fungal virus RnMBV1 nor RnQV1 is known to have the extracellular phase [31]. Hence, no extracellular (re)entry step is expected for their infection cycles. It remains elusive what types of roles these protrusion domains play in the virus infection cycle.

## Surface CrPs atop 3-fold axes

Small-surface trimeric CrPs are present on the RnMBV1 MCPs at the 3-fold axes (Figs 1, 4 and 5). The CrP is composed of C-terminal 135 residues (Lys1292–Gln1426) of the ORF3 in the genome segment dsRNA-2 (Fig 4). In addition, each CrP displays $\alpha$-helix-rich structures, which are formed of seven $\alpha$-helices ($H_A$–$H_G$) (Fig 4B and 4C). The total interface area between each CrP trimer and three MCP_B subunits at the 3-fold axis is 2.6 x $10^3$ $Å^2$. The interactions are intervened by the hydrophobic interaction of the MCP_B loop Pro548-Lys555 and the CrPs and by one hydrogen bond between MCP_B Arg533 and CrP Gly1341 (Fig 5B). The CrPs are not present at all 3-fold axes (Figs 6 and 7). After 3D focused classification, one CrP trimer was recovered with highest intensity and was deemed to have 100% occupancy, whereas other CrP trimers were 30% occupancy (Fig 6). The averaged CrP trimer numbers per particle are 6.7 for full particles and 8.0 for empty particles, respectively (Fig 7). These results probably reflect their limited and weak interactions with MCPs (Fig 5B). Some CrPs might have been lost during purification.

In some dsRNA toti-like viruses that infect multicellular hosts, the surface proteins are weakly interacted with MCPs and are likely involved in their egress and/or entry [23,24]. The

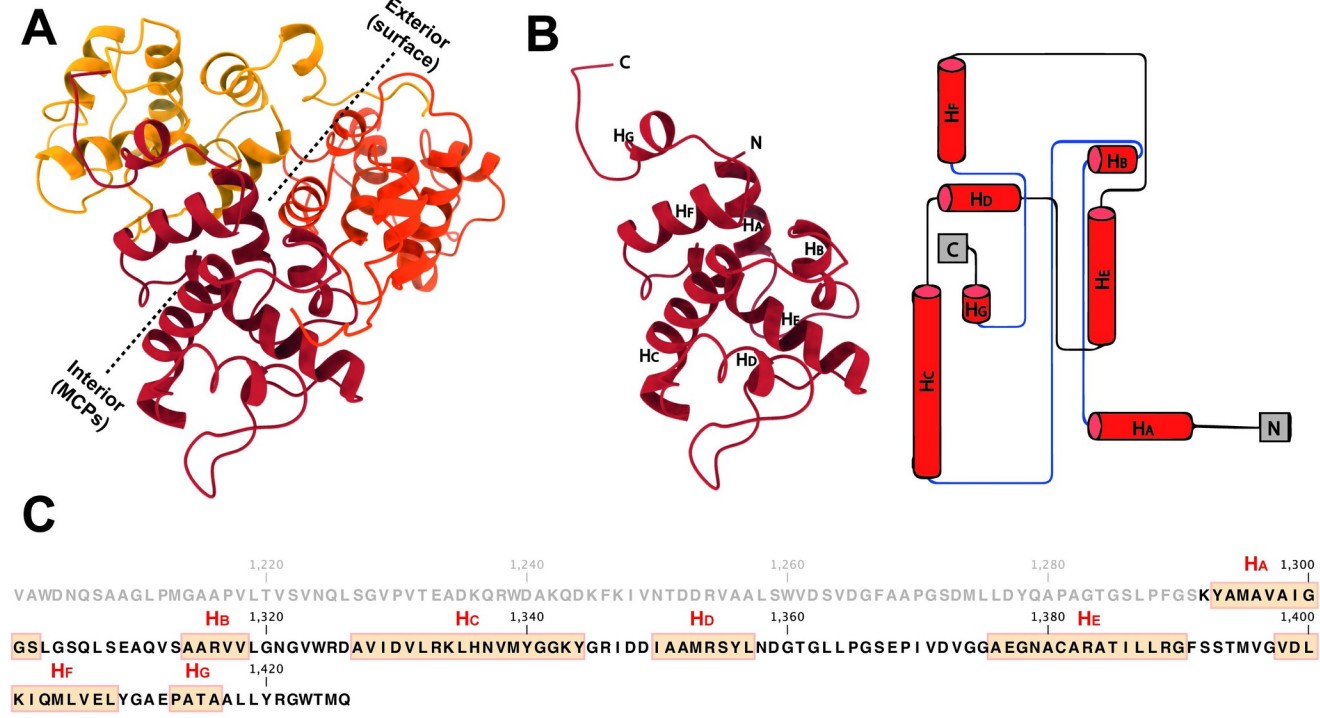

**Fig 4. Atomic model, structural topology, and secondary structural elements of CrP.** A) Atomic model of one CrP trimer (red, orange, and dark yellow). The dashed lines indicate the exterior and interior sides of the CrP trimer. B) Structural topology diagram of the CrP. All $\alpha$-helices are colored red and numbered in order (assigned to be $H_{A-G}$ from the N- to C-terminus). C) Amino acid sequence organization of CrP. The amino acid sequence is a C-terminal portion of RnMBV1 ORF3. The amino acid residues of CrP are marked in black bold from Lys1292 to Gln1426. The orange boxes highlight $\alpha$-helices.

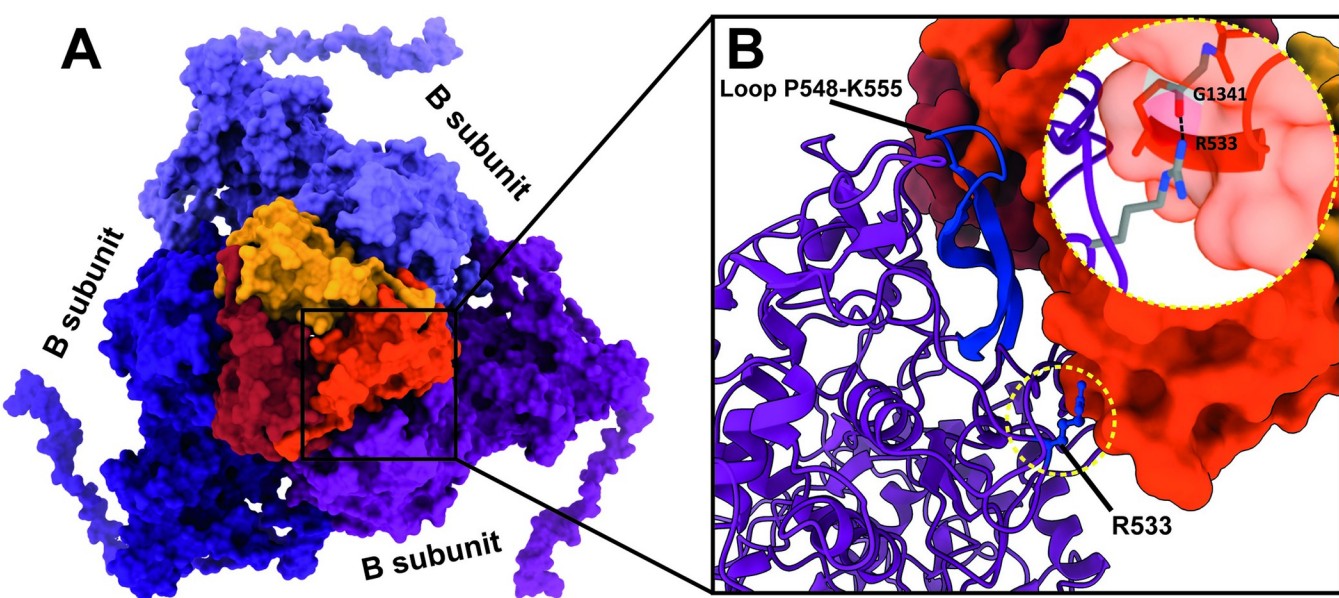

**Fig 5. CrPs and their interactions with MCPs.** A) Enlarged top view of the CrP trimer. Each trimer of the CrPs (red, orange, and dark yellow) is located on the top of the MCP_B subunits (purple, light purple, and dark purple) at the 3-fold axis. B) Close-up view of the interactions between CrP trimer and MCP_B subunit. The MCP_B subunit loop P548-K555 (dark blue) interacts with the CrP. The hydrogen bond between residues MCP_B subunit R533 and CrP G1341 is highlighted in a dashed-yellow circle.

CrP also reminds us about the assembly of the outer shell for the transmission in sedoreoviruses, whose intermediates only comprise outer capsid proteins at 3-fold axes [46]. However, there is no sequence or structure similarity between the RnMBV1 CrPs and these surface proteins. The CrPs might still serve as a factor together with the protrusion domain in MCPs for facilitating transmissions. The weak binding between the CrPs trimer and MCPs indicates that the CrPs possibly function in transmissions even without the full occupancy on the capsid surface (Figs 6 and 7). Most field-collected megabirnaviruses have two genomic segments, while some lab strains are viable without dsRNA-2 or dsRNA-2-encoded proteins [35,36,47]. Namely, virion transfection resulted in the rearrangement of RnMBV1, leading to the inability to express ORF3 and ORF4, or loss of dsRNA-2 in RnMBV2 and SsMBV1 [35,36]. However, the conservation of hypothetical ORF3 showing approximately 54% amino acid sequence identity between RnMBV1 and RnMBV2 suggests its important role. One possibility is the involvement of CrPs in megabirnavirus lateral transmission. Mycoviruses in general can be transmitted horizontally via hyphal fusion between vegetatively compatible strains of a single species or between related species [38,48–50]. However, flies and plants were recently shown to serve as vectors for viruses infecting phytopathogenic ascomycetes and basidiomycetes, respectively [51,52]. In fact, fungal viruses are horizontally transferred to *R. necatrix* possibly from unidentified organisms in the soil [53,54], implying the presence of vector organisms. It is, therefore, of interest to speculate that the CrPs are associated with horizontal virus transmission under natural conditions.

## Pore obstruction

For almost all non-enveloped *T* = 1 icosahedral dsRNA viruses, the pores on the viral capsid are essential for the uptake of nucleoside triphosphates (NTPs) from the hosts and the release of nascent transcribed gene [8,10,18,29,55,56]. Additionally, in most cases, the pores are

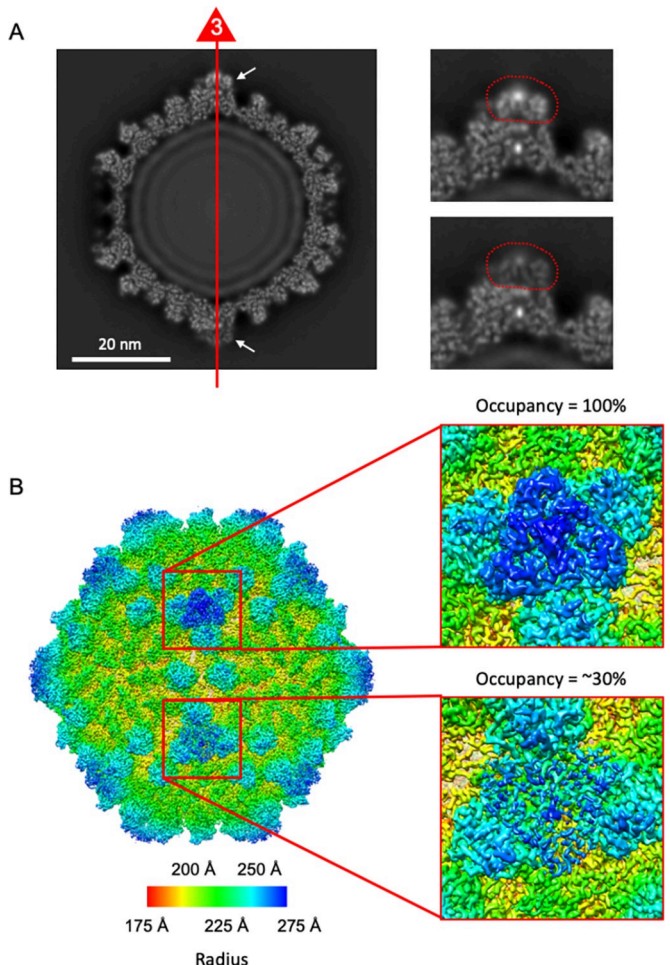

**Fig 6. Reconstructed cryo-EM map after 3D focused classification on CrP trimers.** A) Overall central cross-section of the RnMBV1 cryo-EM map viewed along the icosahedral 3-fold axis (left panel). Enlarged views of the CrPs (right panels). The CrP trimer is surrounded by a red dotted line. The map intensity level in one CrP trimer (lower right panel) is weaker than that in the other regions, while the map intensity in another CrP trimer (upper right panel) is as strong as that in the other regions. B) Surface representation of reconstructed cryo-EM map. The intensity level in one CrP trimer is recovered as high after the 3D focused classification, which is deemed to be 100% occupancy (right upper panel). The map intensity in the other CrP trimers is low, which corresponds to approximately 30% occupancy (right bottom panel). The image is colored radially according to the distance from particle center.

located at the 5-fold axes [8,10,55–57], however, 2/3-fold pores have also been proposed in RnQV1 and victorivirus [40,58]. Hence, for *Megabirnaviridae* viruses, the pores are also expected to exist; however, the 5-fold pore is obstructed in the RnMBV1 capsid (Fig 8A and 8B). Positively charged residues Arg140 and Arg141 conformationally block the pore and contribute to positive surface charges (Fig 8B and 8C). A positively charged cluster can enhance the NTPs uptake, as described previously for HIV [59] and dsRNA toti-like viruses [8]. The C-terminal arm (Val1198–Gln1203) of each MCP_B subunit also contributes to the pore configurations interiorly (Fig 8D), which is unique for the *Megabirnaviridae* RnMBV1 capsid. The exterior and interior sides of the pore become obstructed/narrower because of the Arg residue obstructions and the C-terminal arms; however, the pore still seems to retain approximately 1 nm, on average, to enable release of their synthesized ssRNA gene (Fig 8E).

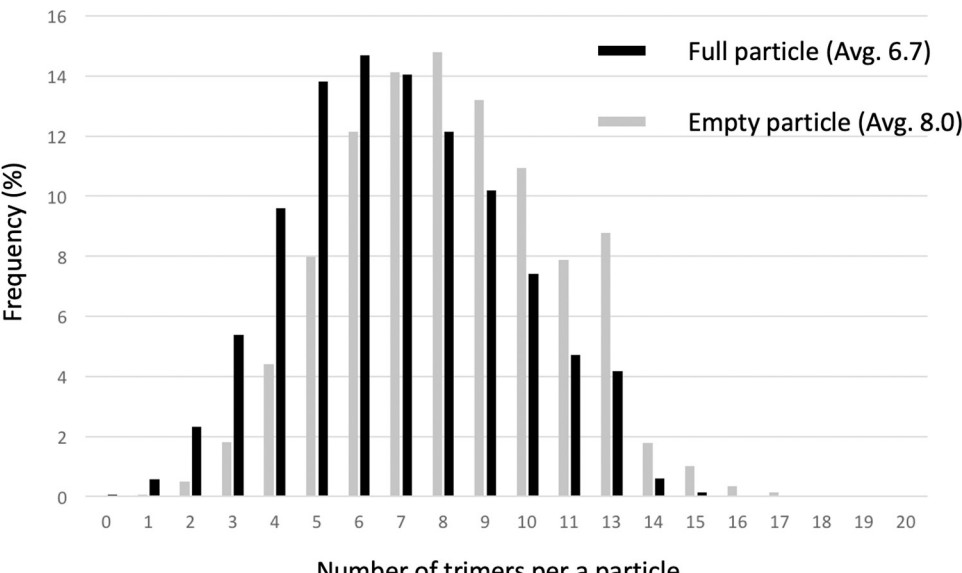

**Fig 7. Number of CrP trimers on the particle and their distribution.** After the 3D focused classification of CrP trimers, the number of CrP trimers bound to the particle is counted and the distribution is plotted. The averaged numbers of presented CrP trimers per particle are 6.7 and 8.0 for full and empty particles, respectively.

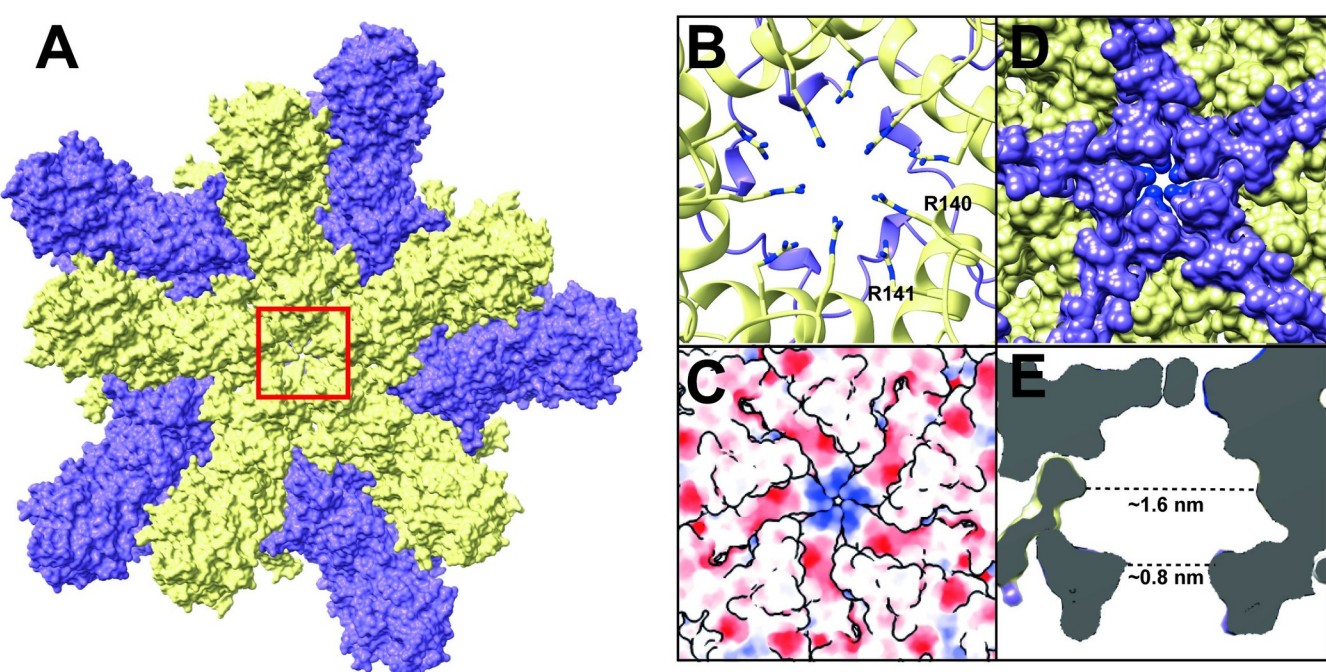

**Fig 8. Structure of an obstructed pore and its properties.** A) Top view of the 5-fold obstructed pore. MCP_A and B subunits are colored yellow and purple, respectively. B) Close-up view (front view). The obstructing R140 and R141 residues of the MCP_A subunits are shown as stick representation. C) Electrostatics surface of the 5-fold pore. D) Backside view of the 5-fold pore. The C-terminal arm of the MCP_B subunits is colored purple. E) Cross-section of the obstructed pore. The dashed lines indicate the calculated pore size in diameter.

## Materials and methods

### Sample preparation

RnMBV1 particles were purified according to Chiba *et al.*'s [32] method. The *R. necatrix* W779 strain was grown at 25˚C in Difco potato dextrose broth (PDB, Becton Dickinson, Sparks, MD, USA) in the dark. Approximately 35 g (wet weight) of the mycelia was harvested and ground into powder in the presence of liquid nitrogen. The virus particles were then extracted from the homogenates in 150 mL of 100 mM sodium phosphate, pH 7.0, through clarification with Vertrel XF (Du Pont-Mitsui Fluorochemicals Co., Tokyo) for potato dextrose agar (PDA)-cellophane cultures. After the addition of NaCl and PEG 6000 to the final concentrations of 1% (w/v) and 8% (w/v), respectively, the particles were pelleted by high-speed centrifugation. After an additional round of differential centrifugation, the RnMBV1 particles were purified through sucrose gradient 10%–40% (w/v) centrifugation at 70, 000 x *g* for 2 h. The virus-containing fractions were pelleted and resuspended in 100 μL of 50 mM sodium-phosphate buffer (pH 7.0).

### Cryo-EM data acquisition and image analysis

For the cryo-EM data collection, 2–3 μL of sample solution was applied to a holey carbon grid (Quantifoil R2/1, Mo 300 mesh; Quantifoil Micro Tools GmbH) covered with a thin amorphous carbon film at 4˚C with 100% humidity. After waiting for 30 s, the excess sample solution present on the cryo-EM grids was blotted with filter papers and then, these grids were plunge-frozen into liquid ethane using a Vitrobot Mark IV (Thermo Fisher Scientific). The EM grids were examined with a 300-kV Titan Krios cryo-electron microscope (Thermo Fisher Scientific) incorporating a field emission gun and a Cs-corrector (CEOS GmbH). Cryo-EM movies were recorded at a nominal magnification of x 59,000 using a Falcon 3EC direct electron detector (calibrated pixel size of 1.12 Å) (Thermo Fisher Scientific). The nominal defocus range was –1.00 to –2.75 μm. Each exposure of 48 electrons/Å$^2$ for 2.0 s was dose-fractionated into 39 frames. The cryo-EM data collection is summarized in S1 Table. The three-dimensional (3D) capsid structure of RnMBV1 was reconstructed using RELION 3.0 [60]; the procedure of the structural analysis is summarized in S1 Fig. The movie frames were aligned and summed into a dose-weighted image using MotionCor2 software [61], and the contrast transfer function (CTF) parameters were estimated using the CTFFIND4 program [62]. The micrographs exhibiting poor power spectra (based on the extent of Thon rings) were rejected (4.5 Å resolution cutoff). To determine the 3D model of RnMBV1, 45,869 particles were automatically picked from 2,734 micrographs and then used for reference-free two-dimensional (2D) classification. Then, 39,272 particles were selected from good 2D classes (S1E Fig) and subjected to 3D classification with an icosahedral symmetry. After 3D classification, two good classes appeared (S1A Fig). The particles in class III (12,230 particles) were filled with the genome (full particles), while those in class II (14,602 particles) lacked the genome (empty class). We selected particles in the good full- and empty-particle classes separately and used them for further structural analyses. The 3D refinement and post-processing, including CTF refinement and Bayesian polishing, yielded maps of both full and empty particles at 3.2 Å resolution, which were estimated by the gold-standard Fourier shell correlation at 0.143 criterion [63, 64] (S1F Fig). To determine the CrP structure, we performed focused 3D classification of the CrP trimers using a mask covering the icosahedral capsid map (S1C Fig) after the particle orientations were expanded with an icosahedral symmetry. The particle orientations in a good 3D class were selected and used for further structural 3D refinement and post-processing. The final map after the focused classification was reconstructed from 244,609 particle orientations

at 3.3 Å resolution (Figs 6, S1B and S1F). The number of the CrP trimers bound to one virion particle was counted based on the classified particle orientations (Fig 7).

## Atomic modeling and refinement

The atomic model of the CPs was first manually built in the cryo-EM map of the full particles with applied icosahedral symmetry (S1A Fig), and that of the CrPs was built in the model of symmetry-expanded asymmetric reconstruction (Figs 6 and S1B) using Coot 0.8.9.2 [65]. The refinement of the obtained models was iteratively performed using PHENIX 1.15 [66], and then, the refined atomic models were further corrected manually using Coot. The validation statistics of the final atomic models are shown in S1 Table.

## Structural analysis

The viral CP structures akin to that of RnMBV1 were searched against all registered structures in the PDB database by the Dali online server [67]. In total, six dsRNA virus CP structures were obtained. These six CP structures and the RnMBV1 MCP structures were applied to pairwise structure-based alignments using the MUSTANG program [68]. Then, the all-to-all Root Mean Square Deviation (RMSD) values were thoroughly calculated in the aligned structures and used for generating structure phylogeny, as previously described [69, 70], using a neighbor-joining method [71] in MEGA X [72]. The aligned structures were also used to display conserved and unique protrusion domains of RnMBV1. The protein structure diagrams of the MCPs and CrP were generated by Pro-origami [73] and further modified by Inkscape. To prepare the figures, the UCSF Chimera and ChimeraX were used [74, 75]. The buried areas (buried solvent accessible surface areas, BaveSAS) were calculated in UCSF Chimera [74].

## Supporting information

**S1 Table. Cryo-EM data collection, refinement, and validation statistics.**
(PNG)

**S1 Fig. Summary of the cryo-EM data analysis.** A) Flowchart of the RnMBV1 MCP data analyses of the cryo-EM micrographs using RELION. B) Flowchart on generating the RnMBV1 CrP 3D reconstruction using RELION. C) CrP's location on the RnMBV1 capsid on the map. D) Cryo-EM raw image of the RnMBV1 particles (empty/full). E) 2D class overview. F) Gold-standard FSC resolution curves of the full particle in icosahedral (I) symmetry (solid black line), empty particle in symmetry I (dashed line), and full particle in C1 symmetry (gray line) reconstructions for RnMBV1 virions. The resolutions of these cryo-EM models are estimated as 3.2, 3.2, and 3.3 Å, respectively (FSC cutoff = 0.143). G) Cryo-EM map and fitting with the atomic models of the protrusion domain (yellow and orange shown in backbone representation only), the C-terminal arms of the MCPs (in ball-and-stick mode), and the CrPs trimer (red, orange, and dark yellow shown in backbone representation). H) Overall 3D reconstructions. Top panel: The overall capsid map with or without the presence of CrP reconstructions. Bottom panel: The back half of the capsid 3D reconstructions shows the interior features with or without the presence of CrP occupancy. The capsid is colored from blue to red, in accordance with the estimated local resolutions.
(PNG)

**S2 Fig. Structural alignments, phylogeny, and superimpositions of the RnMBV1 MCP and the related CP of dsRNA viruses.** A) Summary table of viral CP structures similar to that of the RnMBV1 MCP identified using Dali search. The background colors indicate different taxonomy clades. B) Structure-based phylogenetic tree of RnMBV1 MCP and structurally close-

related six CPs. C) Superimposition of RnMBV1 MCP and RnQV1 CP. D) Superimposition of RnMBV1 MCP, RnQV1 CP, and yeast ScV-L-A CP. Dashed circles highlight protrusion domains of RnMBV1 and RnQV1.
(PNG)

**S3 Fig. Atomic model, structural topology, and secondary structural elements of MCP.** A) Atomic model of one MCP colored by domains. The protrusion domains (amino acid residues 433–648 and 965–1125) are colored yellow and orange, respectively and the conserved domains (amino acid residues 11–432 and 649–964) are colored dark gray and light gray. The C-terminal arm (amino acid residues 1126–1236) is colored dark blue. B) Structural topology diagram of the MCP. The color codes correspond to those in A). The α-helices and β-strands are shown as red cylinders and light blue arrows and named accordingly. The blue, translucent, rounded boxes cover the regions containing β-sheets. C) Amino acid sequence organization of the MCP. The orange and green boxes highlight α-helices and β-strands, respectively. The colors and labels correspond to those of B).
(PNG)

## Acknowledgments

The authors are grateful to Dr. Satoko Kanematsu for her generous gift of the fungal strain W779 of *R. necatrix*.

## Author Contributions

**Conceptualization:** Naoyuki Miyazaki, Nobuhiro Suzuki.

**Data curation:** Naoyuki Miyazaki.

**Formal analysis:** Han Wang, Naoyuki Miyazaki, Kenta Okamoto.

**Funding acquisition:** Naoyuki Miyazaki, Nobuhiro Suzuki, Kenta Okamoto.

**Investigation:** Naoyuki Miyazaki.

**Methodology:** Naoyuki Miyazaki, Kenta Okamoto.

**Project administration:** Naoyuki Miyazaki, Nobuhiro Suzuki, Kenta Okamoto.

**Resources:** Lakha Salaipeth, Naoyuki Miyazaki, Nobuhiro Suzuki, Kenta Okamoto.

**Software:** Han Wang, Naoyuki Miyazaki, Kenta Okamoto.

**Supervision:** Naoyuki Miyazaki, Nobuhiro Suzuki, Kenta Okamoto.

**Validation:** Han Wang, Naoyuki Miyazaki, Kenta Okamoto.

**Visualization:** Han Wang, Naoyuki Miyazaki, Kenta Okamoto.

**Writing – original draft:** Han Wang, Naoyuki Miyazaki, Nobuhiro Suzuki, Kenta Okamoto.

**Writing – review & editing:** Han Wang, Naoyuki Miyazaki, Nobuhiro Suzuki, Kenta Okamoto.

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
