## [Decision Letter · Decision Letter 0]

29 Nov 2022

Dear Dr. Wang,

Thank you very much for submitting your manuscript "Capsid structure of a metazoan fungal dsRNA megabirnavirus reveals its uniquely acquired structures" for consideration at PLOS Pathogens. As with all papers reviewed by the journal, your manuscript was reviewed by members of the editorial board and by several independent reviewers. As you will see, the reviewers appreciated the novelty of the organization of this particular dsRNA fungal virus. Based on the reviews, we are likely to accept this manuscript for publication, providing that you modify the manuscript according to the review recommendations.

Sincerely,

Félix A. Rey

Academic Editor

PLOS Pathogens

Guangxiang Luo

Section Editor

PLOS Pathogens

Kasturi Haldar

Editor-in-Chief

PLOS Pathogens

orcid.org/0000-0001-5065-158X

Michael Malim

Editor-in-Chief

PLOS Pathogens

orcid.org/0000-0002-7699-2064

Reviewer Comments (if any, and for reference):

Reviewer's Responses to Questions

**Part I - Summary**

Reviewer #1: This manuscript presents a high-resolution structure of a megabirnavirus building up on an earlier low resolution structure. It identifies molecular structures that are proposed to be unique to this virus. This structural study describes the first molecular structure for this viral family and will be of interest to the field.

However, the structure description and figures need to be improved as described in Part III of the review.

Reviewer #2: The manuscript by Wang et al. describes the cryoEM-based atomic model (3.2-Å resolution) of Rosellinia necatrix megabirnavirus 1-W779, which is a nonenveloped icosahedral dsRNA virus that infects the ascomycete fungus Rosellinia necatrix, a causative agent of the lethal plant disease white root rot.

Basic features of the capsid structure formed by the dsRNA1-encoded major capsid protein (MCP) are not unexpected for a dsRNA virus with a 120-subunit T=1 (so-called "T=2") protein shell. There is an especially long C-terminal extension that makes multiple contacts across different subunits, but such terminal extensions, though often not as long, have been seen in other "T=2" capsids. The protrusion domain, whose function remains unknown, appears unique, but having a unique (e.g., species- or genus-specific) protrusion domain appears to be the rule rather than the exception among dsRNA viruses with "T=2" capsids.

Probably the most interesting observation is the presence of an unexpected protein trimer anchored atop the 3-fold axes and attributable to the dsRNA3-encoded protein, about which little was previously known but which is now designated the crown protein (CrP). The CrP trimer is apparently not present at every 3-fold axis based on density/occupancy analyses. The authors argue for a role of this protein in horizontal transmission of this virus.

Given interesting aspects of the CrP, I suggest the following additions to the text:

a. The reduced occupancy of the CrP is given very little attention in the text, in fact a single sentence in Results and Discussion: "The CrPs are not present at all 3-fold axes (Figs. S2 and S3), which is probably due to their limited and weak interactions with MCPs (Fig. 4B)." I would like to see this given a bit more emphasis, perhaps by including one of the supplementary figures in the main article and summarizing the nature of the analyses in Results and Discussion. I also think the authors should acknowledge the somewhat trivial explanation that some of the CrP trimers might have been lost during purification.

b. The authors state "The weak binding between the CrPs trimer and MCPs, however, indicates that the CrPs do not function crucially in the [virus-to-host] transmissions." I don't consider this a logically necessary or even probable conclusion. Full occupancy of this protein would not necessarily be required for playing a crucial role in whatever its function might be.

c. The authors state "Fungal viruses are horizontally transferred to R. necatrix from unidentified organisms in the soil (49,50), implying the presence of vector organisms. It is of interest to speculate that the CrPs are associated with horizontal virus transmission under natural conditions." I agree that this seems like an interesting possibility worth discussing, but I feel this idea comes out of the blue. There is no preceding mention of vectors in this manuscript. I think the authors should explain this concept a bit more clearly and how the observations in refs. 49 and 50 might point to it.

Final comment, Title: I think the title could be improved. The phrase "uniquely acquired structures" doesn't tell us much, and "acquired" from where? Also, is the term "metazoan" really needed?

**Part II – Major Issues: Key Experiments Required for Acceptance**

Reviewer #1: Validation reports need to be submitted as obtained from the EMDB and the PDB.

CC mask is 0 for the C1 reconstruction…

Reviewer #2: None to note

**Part III – Minor Issues: Editorial and Data Presentation Modifications**

Reviewer #1: It is important that the fold of the MCP and CrP are introduced in the first couple of figures. A cartoon representation and box diagram of the domains is essential for readability (e.g. bring Fig. S6 into the main text).

In addition, the role of the C-terminal arm is interesting but it is hard to follow the interaction network on individual subunits. Could this be represented over half a capsid? The labels are too small and unreadable even when zooming in.

Fig S4 is also impossible to follow for panels B, D, E because too many structures are superimposed. Consider using a single example or a stereo-image.

Title: What is a “metazoan” fungal virus? P.12, line 212 suggests that RnMBV1 is a metazoan virus, which is confusing.

Abstract: speculations about genome packaging are not supported or discussed in the main text and should be removed.

Detailed comments: text editing recommended. Frequent typos (e.g. p. 10 Megabirnabiridae).

p.5, l.81: clarify why the protruding proteins are “unexpected”.

p.8: what does the RMSD represent (all atoms, main chain…)?

p.8, l. 164: Is this “deletion” a removal of C-terminus in silico?

p.9 l.178: edit “dimerically”

p.9 l.190: Is the C-terminus really expected to play a role in replication during assembly as suggested here without further evidence? This needs to be clarified or edited.

p. 11: “Extra-surface CrPs”. Please reword.

p. 11, l.233: “CrP of RnMBV1 can be associated with the protusion protein/fiber…”. I’m not clear about what’s meant here. Is CrP analogous to the fiber or physically associated? Why would it be involved in egress and/or entry for an intracellular virus?

Reviewer #2: None to note except as noted in Part I.

PLOS authors have the option to publish the peer review history of their article (what does this mean?). If published, this will include your full peer review and any attached files.

Reviewer #1: No

Reviewer #2: No

Figure Files:

Data Requirements:

Reproducibility:

References:

---

## [Editor Report · Decision Letter 1]

15 Jan 2023

Dear Dr. Wang,

Thank you very much for submitting your manuscript "Capsid structure of a fungal dsRNA megabirnavirus reveals its previously unidentified surface architecture" for consideration at PLOS Pathogens. As with all papers reviewed by the journal, your manuscript was reviewed by members of the editorial board, which found that the manuscript according to the reviewer's comments, addressing all the issues they raised, resulting in significant improvement. The paper describes the structure of a double-stranded RNA (dsRNA) virus that kills a fungal pathogen of plants, and which could have important biotechnological applications. Furthermore, compared to other dsRNA viruses of the same family, this particular virus has an additional "crown" protein, which contributes to a novel organization of the virion. The manuscript will therefore be of broad interest, justifying publication in PLoS Pathogens.

the Editorial Board found, however, that the sentence in the abstract: "Contrary to the other structurally associated viral capsid proteins, the RnMBV1 capsid protein structure exhibits an extra-long C-terminal arm and a surface protrusion domain". It is not clear what is ment by  "the other structurally associated viral proteins". Perhaps the authors mean "compared to dsRNA viruses of the same family, the RnMBV1 capsid protein structure has an extra-long C-terminal arm and a surface protrusion"? But perhaps they mean something else. It is important that the abstract is clear. We therefore ask you to reword the abstract, also removing the period in "Rosellinia. necatrix".

Sincerely,

Félix A. Rey

Academic Editor

PLOS Pathogens

Guangxiang Luo

Section Editor

PLOS Pathogens

Kasturi Haldar

Editor-in-Chief

PLOS Pathogens

orcid.org/0000-0001-5065-158X

Michael Malim

Editor-in-Chief

PLOS Pathogens

orcid.org/0000-0002-7699-2064

The authors have revised the manuscript according to the reviewer's comments, which have significantly improved it. It describes the structure of a double-stranded RNA (dsRNA) virus that kills a fungal pathogen of pants, and which could have important biotechnological applications. Furthermore, compared to other dsRNA viruses of the same famil, this particular one has an additional "crown"protein, with a novel organization. The manuscript will therefore be of broad interest, justifying publication in PLoS athogens..

I ask the authors to make one change in their abstract, as the sentence: "Contrary to the other structurally associated viral capsid proteins, the RnMBV1 capsid protein structure exhibits an extra-long C-terminal arm and a surface protrusion domain". It is not clear which "the" other structurally associated viral proteins they mean. Perhaps they mean "compared to dsRNA viruses of the same family, the RnMBV1 capsid protein structure has an extra-long C-terminal arm and a surface protrusion"? But perhaps they mean something else. It is important that the abstract is clear. And since the will be modifying the abstract, they should also remove the period in "Rosellinia. necatrix"

Reviewer Comments (if any, and for reference):

Figure Files:

Data Requirements:

Reproducibility:

References:

---

## [Editor Report · Decision Letter 2]

25 Jan 2023

Dear Dr. Wang,

We are pleased to inform you that your manuscript 'Capsid structure of a fungal dsRNA megabirnavirus reveals its previously unidentified surface architecture' has been provisionally accepted for publication in PLOS Pathogens.

Best regards,

Félix A. Rey

Academic Editor

PLOS Pathogens

Guangxiang Luo

Section Editor

PLOS Pathogens

Kasturi Haldar

Editor-in-Chief

PLOS Pathogens

orcid.org/0000-0001-5065-158X

Michael Malim

Editor-in-Chief

PLOS Pathogens

orcid.org/0000-0002-7699-2064
---

## [Editor Report · Acceptance letter]

22 Feb 2023

Dear Ms Wang,

We are delighted to inform you that your manuscript, "Capsid structure of a fungal dsRNA megabirnavirus reveals its previously unidentified surface architecture," has been formally accepted for publication in PLOS Pathogens.

Best regards,

Kasturi Haldar

Editor-in-Chief

PLOS Pathogens

orcid.org/0000-0001-5065-158X

Michael Malim

Editor-in-Chief

PLOS Pathogens

orcid.org/0000-0002-7699-2064